# A Breakthrough Brought about by Targeting KRAS^G12C^: Nonconformity Is Punished

**DOI:** 10.3390/cancers14020390

**Published:** 2022-01-13

**Authors:** Wenjuan Ning, Zhang Yang, Gregor J. Kocher, Patrick Dorn, Ren-Wang Peng

**Affiliations:** 1Division of General Thoracic Surgery, Inselspital, Bern University Hospital, University of Bern, Murtenstrasse 28, CH3008 Bern, Switzerland; wenjuan.ning@dbmr.unibe.ch (W.N.); zhang.yang@dbmr.unibe.ch (Z.Y.); Gregor.Kocher@insel.ch (G.J.K.); Patrick.Dorn@insel.ch (P.D.); 2Department for BioMedical Research (DBMR), Inselspital, Bern University Hospital, University of Bern, Murtenstrasse 28, CH3008 Bern, Switzerland

**Keywords:** KRAS-mutant cancer, KRAS^G12C^, KRAS^G12C^ inhibitors, acquired resistance, combination therapy

## Abstract

**Simple Summary:**

KRAS is the most common oncogene in human cancers and has long been considered ‘‘undruggable’’—that is, until recently, when covalent inhibitors that selectively target KRAS^G12C^ substitution were developed. The satisfactory results of multicenter clinical trials has led to the recent approval of therapy with KRAS^G12C^ inhibitors. Although KRAS^G12C^ allele-specific drugs have greatly improved the clinical outlook for patients with KRAS^G12C^ tumors, particularly lung adenocarcinomas, in which the KRAS^G12C^ mutant is most prevalent compared with other KRAS mutations, inevitable challenges, such as intrinsic and acquired drug resistance, must be overcome to maximize the efficacy of KRAS^G12C^ inhibitor therapy. Recent studies have shown that compensatory signaling pathways, such as the PI3K/AKT/mTOR pathway, and epigenetic reprogramming, e.g., epithelial-to-mesenchymal transition (EMT), are common mechanisms that mediate intrinsic resistance to KRAS^G12C^ inhibitors, whereas acquired resistance and ensuing recurrent disease can arise when cancer cells acquire secondary mutations in the KRAS protein that impair the covalent binding of KRAS^G12C^ inhibitors. The identification and targeting of KRAS^G12C^ inhibitor resistance mechanisms holds promise for novel strategies to effectively treat patients with KRAS^G12C^-mutant cancers.

**Abstract:**

KRAS is the most frequently mutated oncogene in lung carcinomas, accounting for 25% of total incidence, with half of them being KRAS^G12C^ mutations. In past decades, KRAS enjoyed the notorious reputation of being untargetable—that is, until the advent of G12C inhibitors, which put an end to this legend by covalently targeting the G12C (glycine to cysteine) substitution in the switch-II pocket of the protein, inhibiting the affinity of the mutant KRAS with GTP and subsequently the downstream signaling pathways, such as Raf/MEK/ERK. KRAS^G12C^-selective inhibitors, e.g., the FDA-approved AMG510 and MRTX849, have demonstrated potent clinical efficacy and selectivity in patients with KRAS^G12C^-driven cancers only, which spares other driver KRAS mutations (e.g., G12D/V/S, G13D, and Q61H) and has ushered in an unprecedented breakthrough in the field in recent decades. However, accumulating evidence from preclinical and clinical studies has shown that G12C-targeted therapeutics as single agents are inevitably thwarted by drug resistance, a persistent problem associated with targeted therapies. A promising strategy to optimize G12C inhibitor therapy is combination treatments with other therapeutic agents, the identification of which is empowered by the insightful appreciation of compensatory signaling pathways or evasive mechanisms, such as those that attenuate immune responses. Here, we review recent advances in targeting KRAS^G12C^ and discuss the challenges of KRAS^G12C^ inhibitor therapy, as well as future directions.

## 1. Introduction

RAS proteins are a family of GTPases anchored to the plasma membrane that play an essential role in cellular signaling for proliferation, differentiation, and survival by activating multiple signaling pathways [1]. KRAS is the most frequently mutated member of the RAS family, present in 96% of pancreatic ductal adenocarcinoma (PDAC), 52% of colorectal, 32% of lung carcinomas, and to a lesser extent in a variety of other cancers, with alterations mostly occurring at codon G12, G13, and Q61 [2,3,4,5]. KRAS^G12C^ is the fourth most common substitution of all KRAS mutations and the most frequent mutant isoform in non-small cell lung cancers (NSCLCs), accounting for 41–49% of NSCLC cases with G12 substitutions [5,6,7]. The G12C substitution in KRAS also happens, as has been reported, in 3% of all colorectal cancers and 2% of all pancreatic ductal adenocarcinomas [8,9], while, based on TCGA database, the incidence of G12C substitution in colon cancers reaches 11% (Figure 1).

KRAS proteins cycle between two conformational states—the guanosine diphosphate (GDP)-bound inactive form, and the guanosine triphosphate (GTP)-bound active form that further transduces the activating signals to downstream cascades, such as the mitogen-activated protein kinases (MAPK) pathway and the PI3K/AKT/mTOR signaling pathway [4,10]. In MAPK signaling, RAF is first recruited and activated by the GTP-bound RAS protein, and the activated RAF then activates MEK via phosphorylation, which in turn triggers the next node of the MAPK signaling cascade by phosphorylating and activating ERK [6]. Although it has been shown that PI3K can be regulated by multiple oncogenesis factors, the PI3K/AKT/mTOR axis is considered to be the second most significant pathway downstream of the RAS family [11]. Moreover, RAS can also transduce cellular signals through other signaling effectors, such as RAL GTPases [12].

Interconversion between GDP-bound and GTP-bound RAS is assisted by guanine-nucleotide exchange factors (GEFs) and GTPase-activating proteins (GAPs), as the affinity of RAS to both GDP and GTP is high and the intrinsic GTPase activity of RAS is low [13,14]. As a result, GEFs, such as SOS1/2, replace RAS-bound GDP with GTP and induce a conformational change of RAS, which serves as a conformational entity recognized by downstream effectors and switches on RAS signaling, while GAPs, e.g., neurofibromin 1 (NF1), accelerate the hydrolysis of RAS-bound GTP and bring RAS back to the inactive state to terminate RAS signaling [13,15]. Genetic alterations, such as missense mutations at the codon G12C of KRAS, impair the GTPase activity by disrupting the association of RAS with GAP, which renders RAS in the active status, leading to the hyperactivation of the downstream signaling pathway and uncontrolled cellular proliferation and disordered differentiation [14].

Structural studies (16,17) have shown that RAS proteins have relatively smooth surfaces and no obvious allosteric sites (e.g., hydrophobic pockets/grooves) for drug binding, with the exception of the nucleotide-binding site, which is difficult to target with a competitive analogue due to the high affinity of RAS for both GDP and GTP. In particular, the two dynamic regions involved in effector interactions, switch-I (SWI) and switch-II (SWII), undergo dramatic conformational changes depending on which nucleotide (GDP or GTP) is bound [16,17]. This suggests the possibility of allosterically altering the effector binding of RAS via the switch-I and switch-II regions. However, the identification of such potential binding sites has presented significant difficulties in drug development through structure–activity relationship studies.

Although KRAS was the first oncogene identified in human cancers, and it has been extensively studied to target RAS-mutant cancers by inhibiting the downstream signaling pathways [4,18,19,20], the direct inhibition of RAS was considered to be an insurmountable task until a cysteine-reactive chemical was identified that selectively binds to KRAS^G12C^ but not to other mutant proteins (e.g., KRAS^G12D^, KRAS^Q61H^). In this review, we briefly present the development of these inhibitors, their limitations, and potential solutions.

## 2. Challenges in Therapeutic Targeting of KRAS

The tractability of RAS as a drug target has been significantly confounded for several reasons. First, given the high affinity of GDP-KRAS complexes for GTPs and the abundance of cellular GTPs, it is technically difficult to interfere with the GDP–GTP exchange using GTP analogs that compete for GTP binding [21,22]. The strategy to prevent GTP-KRAS formation by interfering with the interaction with GEFs also failed, as the inhibition affects both KRAS wild-type and mutant cells [16,23]. Finally, RAS proteins lack known allosteric regulation sites, which has thwarted efforts to develop allosteric inhibitors.

Meanwhile, it has been noticed that the surface formation of RAS is highly flexible [16,17]. Based on this observation, recent attempts have been made to identify new binding sites/grooves on the surface of KRAS, which was previously regarded as smooth, and to develop selective inhibitors that bind to the groove.

## 3. G12C Mutation: A Tale of Drugging the Undruggable

The discovery of an ideal groove on the smooth surface of the KRAS protein seemed to be an impossible mission until 2013, when it was shown that the switch-II pocket (S-IIP) is potentially targetable when KRAS proteins have the G12C substitution [24].

In an attempt to overcome the challenges of inhibiting RAS directly, Shokat and co-workers explored a novel approach to covalently target the reactive cysteine-12 of KRAS^G12C^, which is located in close proximity to both the nucleotide pocket and a previously unrecognized surface groove in the switch-II region (S-IIP). Interestingly, the covalent modification of the mutant C12 only occurs in GDP-bound KRAS, as S-IIP is absent in GTP-bound KRAS [10,24]. The formation of an irreversible and covalent bond between the inhibitor and the cysteine residue of KRAS^G12C^ irreversibly disrupts both switch-I and switch-II, reduces the affinity of KRAS for guanosine nucleotides despite the presence of GTP, and allows for the persistent disruption of the adducted protein, which locks KRAS^G12C^ in the GDP-bound conformation. Notably, these inhibitors rely on the intrinsic GTPase of mutant KRAS^G12C^, i.e., the KRAS^G12C^ protein is still able to alternate between their active and inactive state, as GTP-bound KRAS is the predominant form in *KRAS^G12C^*-mutant cancer cells. Such a mutant-specific strategy enables the selective inhibition of KRAS^G12C^ while sparing the other KRAS mutants (e.g., G12D, G12S, G12S, G13D, Q61H, etc.) or *KRAS* wild-type cells (e.g., normal tissue), potentially overcoming the toxicological challenges elicited by the nonselective inhibition of *KRAS*-driven cell growth [24].

**ARS853.** The first G12C-selective inhibitor was developed based on the so-called compound 12, which is described as the most potent candidate in the primary report. ARS853 covalently binds to cysteine residue 12 with high affinity and can signficantly affect the active KRAS protein as well as its downstream effectors, such as the RAF/MEK/ERK and PI3K/AKT/mTOR signaling pathways, by inducing apoptosis and decreasing cell proliferation in vitro [25,26]. The high selectivity of ARS853 has also been demonstrated by the fact that its inhibitory effects can be abrogated by the ectopic expression of KRAS^G12C^ substitution [26].

**ARS1620.** Compared to ARS853, which was successful in in vitro studies, the second-generation inhibitor ARS1620 has improved pharmacological properties (e.g., metabolic stability), making it the first inhibitor with in vivo efficacy [27]. ARS1620 was not only optimized for in vivo stability, but also exhibited an intensive potency in G12C occupation and the effective blockade of RAS signaling [28]. In a Mia.paca-2 cell line xenograft, tumor growth was completely inhibited at a drug dose of 200 mg/kg, and the protein levels of RAS-GTP, phospho-ERK, phospho-AKT, and phospho-S6 were all significantly downregulated. In comparison, ARS1620 shows no effects in KRAS^G12V^ mutant xenografts.

**AMG510 (sotorasib).** Derived from ARS1620, this third-generation sotorasib is the first G12C inhibitor to enter and complete clinical trials. Based on the promising results of the NCT03600883 trial and an ongoing phase III clinical trial (CodeBreaK 200; https://doi.org/10.1016/j.jtho.2020.10.137 accessed on 21 November 2021), AMG510 was recently approved by the FDA (the United States Food and Drug Administration) for the treatment of locally advanced or metastatic NSCLC [29,30]. Compared to ARS853 and ARS1620, a breakthrough of AMG510 is that its binding induces an alternative orientation of histidine 95 (His-95), which provides a novel surface groove on KRAS^G12C^ that can be occupied by the aromatic rings of AMG510, leading to the enhanced interaction of AMG510 with KRAS^G12C^ via van der Waals contacts and the improved potency of AMG510. Indeed, AMG510 exhibits a 10-fold increase in efficacy and improved kinetics compared to ARS1620. The onset of the maximal inhibition of the MAPK pathway occurs 2–4 h after exposure to AMG510, which maintains the inhibition of phospho-ERK in vivo for up to 48 h [29].

The safety and tolerability profile of AMG510 was demonstrated in a phase I study [31] that included 59 cases of NSCLC, 42 cases of colorectal cancer, and 28 cases of other tumors. The total 129 patients were divided into four cohorts with planned doses of 180, 360, 720, and 960 mg per day, respectively. The mean elimination half-life of the drug was 5.5 h. Treatment-related adverse effects occurred in 73 patients (56.58%), of which 15 patients had grade 3 or 4 effects, e.g., an increase in alanine aminotransferase (ALT), diarrhea, anemia, an increase in aspartate aminotransferase (AST), an increase in blood alkaline phosphatase, hepatitis, a decrease in lymphocyte count, an increase in γ-glutamyltransferase, and hyponatremia. The treatment showed high potential anticancer activity, especially in the subgroup of NSCLC: 19 patients had a confirmed partial response and 33 had stable disease. That is, the confirmed response rate was 32.2% and disease control (including objective response and stable disease) was 88.1%. In this subgroup, the median progression-free survival was 6.3 months. In addition, 34 patients with NSCLC in the 960 mg cohort had a more satisfactory disease control rate of 91.2%.

In a phase II study of 126 NSCLC patients, the efficacy of sotorasib was further evaluated [32]. While complete responses were observed in four cases only, a partial response occurred in 42 cases and stable disease in 54 cases. Although the reported overall objective response rate was 37.1% and the disease control rate was 80.6%, the median progression-free survival (PFS) was 6.8 months. Treatment-related adverse events (TRAEs) were observed in 69.8% of patients, with 20.6% of patients experiencing severe TRAEs (twenty-five grade 3 events and one grade 4 event).

**MRTX849 (adagrasib).** MRTX849 is another third-generation drug for targeting KRAS^G12C^ that may well receive final approval. MRTX849 has a relatively long half-life of 17 to 48 h [33,34], and its efficacy in discriminating KRAS^G12C^ activity was widely confirmed in KRAS^G12C^-mutant cell lines under both 2D and 3D conditions. On the contrary, non-KRAS^G12C^-mutant cell lines showed IC_50_ values greater than 1 μM in 2D and greater than 3 μM under 3D conditions [33]. An RNA sequencing analysis of a xenograft mouse model confirmed that ERK-dependent transcription was blocked, accompanied by the reactivation of receptor tyrosine kinase (RTK)- and ERK-dependent signaling.

Based on these in vitro and in vivo data, a phase I/II clinical trial (NCT03785249) was then conducted. Adagrasib is also well tolerated and provides great benefits to patients with KRAS^G12C^-mutant cancers [35,36]. In 110 patients with colorectal cancer or NSCLC, treatment-relative adverse events occurred in 85% of cases, including 33 patients with grade 3 or 4 adverse events. Clinical activity was evaluable in 51 patients, of whom 23 patients had a partial response and 26 patients had stable disease. Overall, the disease control rate was up to 96%. In addition, an objective response rate of 43% was confirmed in 14 phaseI/Ib patients with a longer follow-up.

**Other G12C inhibitors.** GDC-6036, JNJ-74699157 (ARS-3248), and LY3499446 are other inhibitors in the field. The latter two candidates have been enrolled in clinical trials without much reporting of preclinical evidence. The clinical trial of JNJ-74699157 (NCT04006301) was reported to be completed after less than a year, with only 10 participants recruited and no result published. The study of LY3499446 (NCT04165031) was terminated at the end of 2020 due to an unexpected toxicity finding. The clinical trials with KRAS^G12C^ inhibitors are summarized in Table 1.

## 4. Limitations and Possible Approaches to Optimize G12C Inhibitor Therapy

Drug resistance is invariable for targeted therapy, and clinical trials with sotorasib and adagrasib provide compelling evidence for the presence of both intrinsic and acquired drug resistance: in cancer patients with KRAS^G12C^ substitution, most malignant lesions shrink but do not disappear, or have a less-than–20% regression (PR to SD), which eventually relapses within months in every case.

The compensatory activation of bypass signaling pathways is the most common mechanism by which the cytotoxicity of targeted inhibitors is circumvented. The basal level of the KRAS-GTP protein itself is considered to be an indicator of intrinsic resistance to G12C inhibitors [37]. In addition, metabolic and epigenetic factors, the cell cycle checkpoint and immune checkpoint, can also contribute to drug sensitivity and resistance.

The PI3K/AKT/mTOR pathway is an essential effector downstream of the RAS protein, but its activation is not solely dependent on the RAS protein. In addition to the GTP-bound RAS protein, PI3K signaling can also be triggered by receptor tyrosine kinases (RTK) and G protein-coupled receptors (GPCR). The PI3K/AKT/mTOR signaling cascade and RAF/MEK/ERK pathway negatively regulate each other and thus may compensate for each other when mediating cell survival. Since the PI3K/AKT/mTOR and RAF/MEK/ERK pathways regulate each other, the inhibition of MEK can abrogate the suppression of PI3K [38]. Because these two pathways have such a complex interplay, it is not unexpected that combined treatment with KRAS^G12C^ and PI3K inhibitors would have a synergistic effect on KRAS^G12C^-mutant cancer cells. Indeed, Misale S. et al. reported that ARS1620 in combination with various PI3K inhibitors showed synergistic effects both in vitro and in vivo, especially in models intrinsically resistant to ARS1620, such as HCC44, H2122, and SW1573 [39].

Epithelial-to-mesenchymal transition (EMT) is a phenomenon in which epithelial cells lose their polarity and cell–cell adhesion, but gain mesenchymal stem cell properties, such as drug resistance [40]. EMT has been reported to positively correlate with both intrinsic and acquired resistance to AMG510 and ARS1620 in KRAS^G12C^-mutant cell lines [37]. Cell lines resistant to AMG510 have higher EMT scores, and EMT induction through TGFβ treatment renders the otherwise sensitive cell lines resistant to ARS1620 and increases KRAS-GTP protein levels. EMT is also a resource for acquired resistance to G12C inhibitors, with vimentin levels apparently increased in the AMG510-resistant cell lines that were obtained by chronical drug treatment. The molecular basis for EMT-mediated drug resistance could be the activation of the PI3K pathway in mesenchymal-like KRAS^G12C^-mutant cancer cells, or, alternatively, a cell cycle alteration leading to CDK4-dependent growth [41].

Given that SOS1 (GEF of the RAS-GTPase) activates RAS proteins by enhancing the release of GDP from RAS, after which GTP is usually bound, as it is generally found at a higher concentration than GDP in cells, the amount of active SOS1 affects the percentage of GTP-bound KRAS protein. In KRAS-mutant cells, SOS1 inhibitors can reduce p-ERK activity by up to 50%, which has been widely confirmed in a number of KRAS mutant variants [42]. In KRASWT cells, the Ras/MAP kinase pathway can be completely blocked by SOS1 inhibitors, indicating the widespread toxicity of SOS1 inhibition in eukaryote cells [16]. However, the synergistic effects of the SOS1 inhibitor BAY-293 with the first-generation G12C inhibitor ARS853 have been demonstrated [23], as has the SOS1 inhibitor BI-3406 with the MEK inhibitor trametinib [42].

RTKs (e.g., EGFR, FGFR, and VEGFR) act upstream of and heterogeneously impact upon the MAPK signaling pathway, depending on the genetic alternations present [39]. KRAS^G12C^-mutant colorectal cancers (CRC) exhibit high RTK activation and respond relatively poorly to sotorasib, as sotorasib alone barely sustains ERK inhibition. However, cetuximab, a wildly used EGFR inhibitor, has been shown to sensitize sotorasib-refractory CRC to sotorasib. The combination of these two drugs significantly reduces cell viability in vitro and shrinks tumor volumes in PDX mouse models [43]. Similarly, in the xenograft models of the NSCLC cell line H2122 and esophageal carcinoma cell line KYSE-410, adagrasib, in combination with other EGFR inhibitors, such as afatinib, resulted in a desirable outcome [33]. In KRAS^G12C^-mutant lung cancers, a synergistic effect was observed in the dual inhibition of MEK and FGFR1 [44]. First, the MEK inhibitor trametinib was shown to induce greater cell death in FGFR1 knockdown cells. Further, when combined with ponatinib, a pan-RTK inhibitor, trametinib displayed enhanced growth inhibition in a variety of tumor models, including in genetically engineered mice. This combinatorial drug efficacy was accompanied by a pronounced decrease in phospho-ERK, indicating the inhibition of the MAPK pathway by the drug combination. Similarly, the efficacy of the KRAS^G12C^ blockade may also be enhanced by FGFR inhibitors.

SHP2 regulates the MAPK pathway by activating SOS1 and, by phosphorylating SHP2, RTKs relay extracellular signals to the Ras/Raf/MEK/ERK pathway. Therefore, SHP2 inhibition may diminish the upstream signaling from RTK and impair KRAS protein activation, leading to a broad enhancement of the effect of KRAS^G12C^ inhibitors on KRAS^G12C^-mutant cancer cells. Not only was the synergy of ARS1620 and the selective SHP2 inhibitor SHP099 confirmed both in vitro and in vivo, but it has also been shown that the efficacy of KRAS^G12C^ inhibitors can be significantly augmented in KRAS^G12C^-mutant cells via SHP2 knockout [45]. In a murine xenograft model using the cell line KYSE-410 or H358, the combination of adagrasib and the SHP2 inhibitor RMC-4550 resulted in a statistically significant reduction in tumor growth [32]. Based on sufficient theoretical supports and accumulated evidence, a clinical trial (NCT04330664) of the combined treatment of adagrasib and the SHP2 inhibitor TNO155 was launched in the middle of 2020 [46].

The dysregulation of DNA damage repair (DDR) has also been shown to lead to drug resistance. DDR can repair the DNA damage caused by cytotoxic chemicals, such as chemotherapy. In healthy cells, DDR helps to defend against various genotoxic factors, but in cancer cells, increased DDR leads to tolerance to drug treatment [47]. Synergistic effects have been confirmed not only in chemotherapy-induced DNA damages, but also in the combination treatment of PARP and MEK inhibition in ovarian cell lines, as KRAS mutation is associated with the increased sensitivity of MEK inhibitors but a poorer response to PARP inhibitors [48]. This indicates that DNA damage repair pathways are also able to affect intracellular signaling cascades and have the potential to optimize targeted therapy by exploiting the MAPK pathway.

Immune checkpoint inhibitors (ICIs) have attracted a significant amount of attention in recent years, and the PD-1/PD-L1 axis has been shown to interact with the MAPK pathway. PD-L1 is a transmembrane protein encoded by gene CD274 and is a ligand of the PD-1 receptor on active T cells. The PD-1/PD-L1 axis works as an inhibitory checkpoint of activated T cells and promotes peripheral immune tolerance [49]. MAPK signaling activity is able to stabilize CD274 mRNA, and thus enhances PD-L1 expression at the post-transcriptional level [50]. On the contrary, the inhibition of the MAPK pathway prevents EGF- and IFNγ-induced PD-L1 expression through the suppression of CD274 mRNA [51]. Consequently, both sotorasib and adagrasib have been confirmed to augment the efficacy of immunotherapy in immune-competent mice [29,52].

Some of the reported resistance mechanisms and the strategies to maximize the efficacy of KRAS^G12C^ inhibitors are summarized in Figure 2.

## 5. Acquired Resistance and the Strategy to Conquer It

Secondary mutations in KRAS, which interfere with the covalent binding of cysteine 12 residual and G12C inhibitors, are the main cause of acquired resistance [53,54]. Amino acid substitutions, such as A59S, Y96C, and G13D, have been found in both sotorasib- and adagrasib-resistant cells. Independent in vitro and clinical studies have also revealed substitutions of A59T, R68M, and Q61L in sotorasib resistance, and Q99L, R68S, V8E, M72I, G12D/R/V/W, Q61H, H95D/Q/R, and Y96D in adagrasib resistance. Mapping onto the sequence and structure of KRAS proteins, G10 to V14 composes a P-loop in the switch-II pocket, D30 to Y40 composes the switch-I domain, and T58 to M72 composes the switch-II domain [17]. These secondary substitutions weaken or even abolish the efficacy of highly selective G12C inhibitors, leading to drug resistance.

In these cases, broad inhibitions of the MAPK pathway, such as SOS1 and SHP2, may have the potential to overcome resistance and re-sensitize G12C inhibition [37,53,55]. It also works in RTK-mediated resistance. RTK activation upregulates SHP2 phosphorylation and accelerates the progress from GDP-bound KRAS to GTP-bound KRAS, which enables cancer cells to escape the blockade of KRAS^G12C^. Indeed, an analysis of adagrasib-resistant patients with KRAS^G12C^-mutant cancer identified the genomic evolutions of the RTK-RAS signaling pathway [54]. EGFR inhibitors could reverse secondary resistance to sotorasib in colorectal cancers, which, as single agents, barely sustain ERK inhibition, and resistance rapidly develops [43].

PI3K transmits signals from both RAS and RTKs, so the sustained inhibition of the PI3K pathway by G12C inhibitors, such as ARS1620, is unlikely, as shown by phospho-AKT levels [33,39]. As a result, tumors with acquired resistance to G12C inhibitors remain sensitive to the inhibition of the PI3K/AKT/mTOR pathway, and dual inhibition shows better efficacy in resistant cells (Table 2).

## 6. Other Factors Driving Resistance to KRAS^G12C^ Inhibitors

The amount of GTP-bound KRAS protein is considered to be a negative prognostic factor because KRAS^G12C^ inhibitors bind only to cysteine residue 12 in the GDP-bound KRAS protein [37]. Therefore, strategies to downregulate the GTP-bound state and render the KRAS protein readily accessible to G12C inhibitors have been desperately sought. However, the mechanisms regulating the GTP-bound state in cancer cells remain largely unexplored.

RAD51 is reported to be highly expressed in KRAS-mutant cancer cells [56] and is a key component in homologous recombination (HR), a pivotal means to repair double-strand DNA breaks. The upregulation of RAD51 may indicate that the abnormally activated KRAS protein in some way causes increased DNA damage, which in turn leads to the upregulation of HR to repair DNA damage.

Metabolic alterations in glucose, amino acids, fatty acids, or mitochondria also play a potential role in sensitivity and resistance to drugs targeting the MAPK pathway. For example, it has been found that the low expression of a rate-limiting enzyme of glycolytic metabolism, fructose-1,6-bisphosphatase (FBP1), can partially inhibit the activation of ERK1/2 [57]. The activation of ERK1/2, which is associated with poor prognosis in malignant tumors, can re-sensitize gemcitabine-resistant KRAS-mutant PDACs to gemcitabine because the development of resistance to this drug is due to ERK phosphorylation and activation, which can be successfully blocked by FBP1 inhibitors [58].

## 7. Concluding Remarks and Prospective Directions

KRAS^G12C^ inhibitors offer a variety of options for the treatment of cancer with the KRAS^G12C^ mutation and are a milestone in the development of precise medicine. This type of highly potent and selective inhibitor has taken a major step forward and is widely recognized. The accelerated approval given by the FDA is undoubtedly the most important confirmation of the potential of sotorasib and demonstrates the broad enthusiasm for KRAS^G12C^ inhibitors.

Although both sotorasib and adagrasib showed an impressive rate of disease control, we cannot ignore the patients who responded less well or not at all to this type of highly selective inhibition. How they escape this precise targeting is still poorly understood, but we can assume that it depends on the amount of inactive KRAS protein that can be interfered with by other signaling pathways.

Another major problem is acquired resistance, which can be caused by a plethora of different mechanisms. Among others, secondary mutations in KRAS^G12C^ that interfere with the binding of covalent inhibitors with cysteine residue and the compensatory activation of alternative pathways that bypass the blocked KRAS^G12C^ and activate the downstream MAPK pathway are the most common mechanisms that confer extrinsic G12C inhibitor resistance in KRAS^G12C^-mutant cancer cells. 

By reacting covalently with the residue of substituted cysteine 12, occupying the switch-II Pocket and blocking the binding of GTP with KRAS protein, G12C inhibitors were born from an extremely brilliant idea, the introduction of which represents a milestone on the road to targeted therapy—making the impossible possible. They also open a new perspective for targeted therapy—the inhibition of a specific mutation by small molecules. Since the KRAS^G12C^ mutation is widespread in human cancers, we can be confident that G12C inhibitors will raise new hopes, and we can look forward to more inhibitors for precise targets.

## Figures and Tables

**Figure 1 cancers-14-00390-f001:**
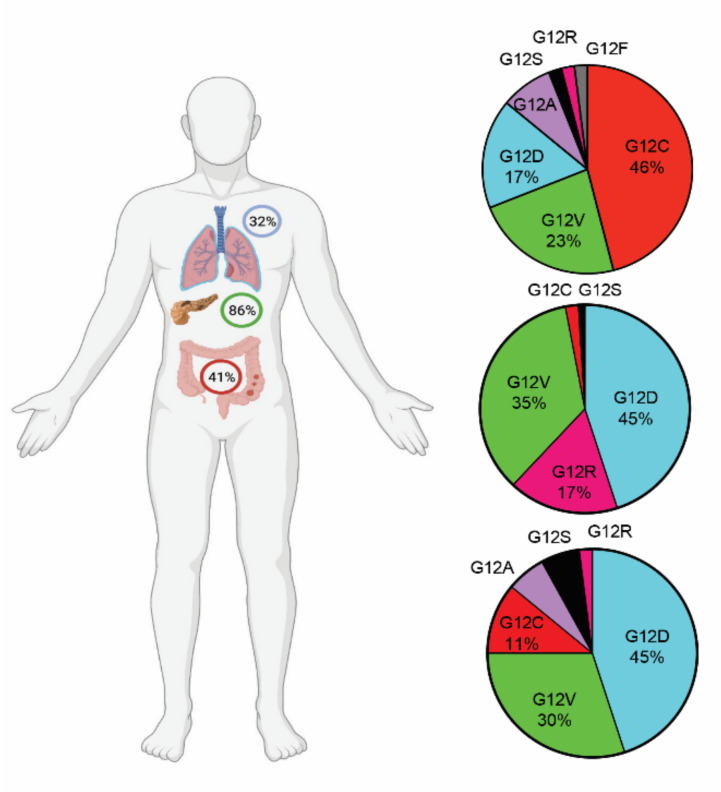
KRAS-G12 substitutions in lung (**top**), pancreatic (**middle**), and colon (**bottom**) cancers. Data are based on patient cohorts in TCGA.

**Figure 2 cancers-14-00390-f002:**
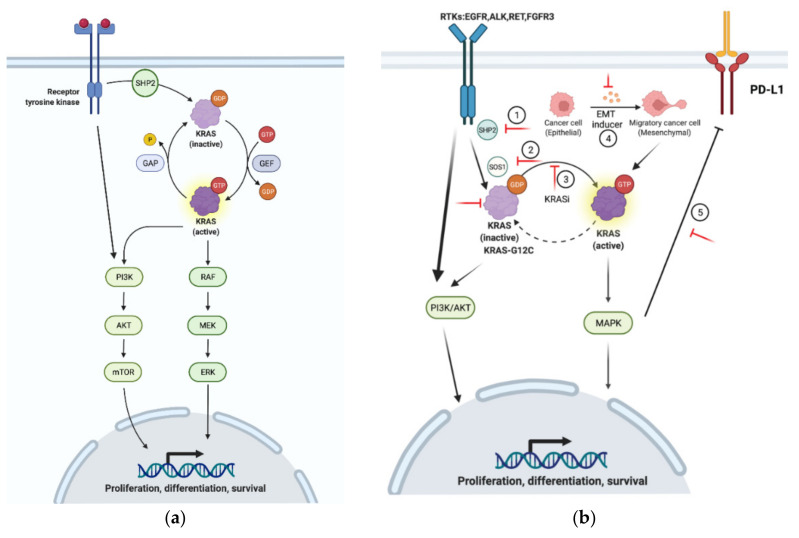
Mechanisms of resistance to KRAS^G12C^ inhibition. (**a**) Under physiological setting, KRAS cycles between active and inactive status. (**b**) Mechanisms of resistance to KRAS^G12C^ inhibitors: (1–3) SHP2, SOS1, and KRAS-GTP activate KRAS and its downstream cascades, which provides positive feedback to RTKs and the PI3K/AKT/mTOR pathway; (4) EMT increases the level of KRAS-GTP and renders KRAS-mutant cells less sensitive to KRAS inhibitors; (5) MAPK promotes PD-L1 expression and leads to the immune evasion of KRAS-mutant cancer cells. The red lines indicate the targets for therapeutic intervention to overcome KRAS^G12C^ inhibitor resistance.

**Table 1 cancers-14-00390-t001:** KRAS^G12C^ inhibitors in clinical trials.

Agent	Strategy	Study Phase	Enrolled Patients	Trial Number
GDC-6036	Combined with atezolizumab (ICI); cetuximab (EGFRi); bevacizumab (VEGFi); erlotinib (EGFRi)	Phase I	KRAS^G12C^-mutant advanced/metastatic solid tumors	NCT04449874
JNJ-74699157 *	Standard treatment	Phase I	KRAS^G12C^-mutant advanced/metastatic solid tumors	NCT040063301
D-1553	Standard treatment	Phase I	KRAS^G12C^-mutant advanced/metastatic solid tumors	NCT04585035
LY33499446 *	Combined with abemaciclib (CDK4/6i); erlotinib; docetaxel	Phase I, II	KRAS^G12C^-mutant advanced/metastatic solid tumors	NCT04006301
Adagrasib	Combined with docetaxel; pembrolizumab (ICI); cizumab (VEGFi); erlotinib	Phase I, II, III	KRAS^G12C^-mutant advanced/metastatic solid tumors	NCT04685135; NCT03785249; NCT04330664; NCT04793958
Sotorasib	Standard treatment	Phase I, II	KRAS^G12C^-mutant advanced/metastatic solid tumors	NCT04380753; NCT04625647; NCT04667234; NCT04933695
Sotorasib	Combined with docetaxel; pembrolizumab; panitumumab (EGFRi)	Phase I, II, III	KRAS^G12C^-mutant advanced/metastatic solid tumors	NCT04303780; NCT03600883; NCT04613596; NCT04185883

* Trials terminated.

**Table 2 cancers-14-00390-t002:** Mechanisms of acquired resistance to KRAS ^G12C^ inhibitors.

Inhibitor	Mechanisms of Resistance	Reference
ARS1620	PI3K pathway activation	[39]
Sotorasib (AMG510)	G13D, A59S/T, Q99L, Y96D/S, R68M, SOS1, SHP2	[53]
	Epithelial-to-mesenchymal transition	[37]
	EGFR co-activation	[43]
Adagrasib (MRTX849)	CDKN2A, RB1 CDK4, CDK6	[33]
	Acquired mutations in KRAS (G12D/R/V/W, G13D, Q61H, R68S, H95D/Q/R, Y96C), high-level amplification of KRAS^G12C^ allele. MET amplification; activating mutations in NRAS, BRAF, MAP2K1, and RET; oncogenic fusions involving ALK, RET, BRAF, RAF1, and FGFR3; and loss-of-function mutations in NF1 and PTEN.	[54]

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
