# Peer review of "A Breakthrough Brought about by Targeting KRASG12C: Nonconformity Is Punished"

_cancers, 2022, doi:10.3390/cancers14020390_

Round 1

Reviewer 1 Report

The authors present an article outlining the challenges and approaches with respect to targeting oncogenic KRAS with a G12C mutation. The authors have touched upon various aspects.

A few suggestions:

  1. The structure/groove/ Switch in the KRASmt is mentioned in a dispersed manner at several places and is a key aspect of this article. It would be helpful if the authors describe the structure of KRAS at the beginning of the article. It would be helpful in explaining as to how G12C mutation is different from other mutations like G12D or G12V.
  2. Please include the limitations of the test drugs, e.g. ARS853, ARS1620
  3. Line 84: Please explain targeting the groove of the “smooth surface of KRAS”
  4. Line 94: Please expand on how the cysteine-reactive small molecules specifically target/bind the substituted cysteine. A pictorial representation would be helpful.
  5. Line 114: A phase III trial is underway for Sotorasib
  6. Line 212: Please introduce RTKs

Minor comments:

Please fix the typographical errors:

  1. Line 62: Second word; KARS change to KRAS

Line 209: “fist-generation”

Reviewer 2 Report

This is a mostly well-written and informative review of a very recent and fast-moving topic.  Suggested edits are below -- these are mostly editorial with a few more substantive comments.

61 (line number) - KRAS misspelled

75 - G12C instead of Glycine12cysteine

117 - This needs some context -- what are the competing molecules?  I assume we are talking about H95 in KRAS, since the referenced paper talks about this interaction but most readers will not be automatically familiar with the significance of this residue.

150 - Indicate that this is a phase 1/2 trial

168 - targeted instead of targeting

203-205 - SOS1 does not phosphorylate GDP but enhances the release of GDP by RAS, after which GTP is usually bound as it is generally found at a higher concentration than GDP in cells

255 - remove "revealed"

256 - "improve PD-L1 ex-pression on post-transcriptional level" would be better as "enhance PD-L1 expression at the post-transcriptional level"

Table 2 - This is a messy table with three entries for sotorasib (referred to in non-consecutive entries as AMG510) -- these should be all be together, all the mechanisms for MRTX849 (which it would be helpful to also label "adagrasib") should be together.

318 - should be inhibitor not inhibitors

331 - remove "of"

327-333 - This paragraph is a little odd.  It is not correct that "only" two types of genetic alterations can lead to G12C inhibitor resistance.  Mutations in the target molecule and bypass mutations are the two general mechanisms we think of when we consider acquired resistance to targeted therapy, but it seems that there are more complex and less well-understood mechanisms as well, as is true for other targeted therapies that do not seem obviously related to the MAPK pathways or the drug-target interaction (EMT, and some of the other mechanisms shown in Table 2).

Reviewer 3 Report

The authors presented the paper "A breakthrough brought about by targeting KRASG12C: nonconformity gets punished"

It will be very interesting to see not only the drug effect but their structure and clear target or ideas of this target. The results will be good to summarize in the picture and/or in the Table.

I see that you review about G12C mutation. However, from the literature and your picture 1, KRAS G12D is one of the major mutations too. Also, you haven't mentioned codon 13 and others. I think you have enlarged this part of the introduction to show the importance and relevance of the G12C.

Reviewer 4 Report

This minireview discusses the possibility of targeting KRAS G12C mutant cancers with KRAS G12C inhibitors and sheds light on the various challenges associated with this targeted therapy and the probable ways to overcome them. The authors should consider the following suggestions for improving the quality of this manuscript before it may be considered for publication. 

  1. The catchphrase "nonconformity gets punished" in the title does not get clearly communicated in the manuscript text. The readers would find it difficult to relate to the subject unless explicitly elaborated.
  2. The authors are advised to revise the manuscript for typos and grammatical errors. 
  3. In Fig 2, MEK has been mislabelled as MERK.
  4. The sentence "Similarly..... outcomes" (line 218-221) is misleading as the referenced article does not involve sotorasib and adagrasib is not an EGFR inhibitor.
